# Disturbance Characteristics of 1T DRAM Arrays Consisting of Feedback Field-Effect Transistors

**DOI:** 10.3390/mi14061138

**Published:** 2023-05-28

**Authors:** Juhee Jeon, Kyoungah Cho, Sangsig Kim

**Affiliations:** Department of Electrical Engineering, Korea University, 145 Anam-ro, Seoul 02841, Republic of Korea; isdf35@korea.ac.kr (J.J.); chochem@korea.ac.kr (K.C.)

**Keywords:** one-transistor dynamic random-access memory, memory array, silicon nanowire, feedback field-effect transistor, positive feedback mechanism

## Abstract

Challenges in scaling dynamic random-access memory (DRAM) have become a crucial problem for implementing high-density and high-performance memory devices. Feedback field-effect transistors (FBFETs) have great potential to overcome the scaling challenges because of their one-transistor (1T) memory behaviors with a capacitorless structure. Although FBFETs have been studied as 1T memory devices, the reliability in an array must be evaluated. Cell reliability is closely related to device malfunction. Hence, in this study, we propose a 1T DRAM consisting of an FBFET with a p^+^–n–p–n^+^ silicon nanowire and investigate the memory operation and disturbance in a 3 × 3 array structure through mixed-mode simulations. The 1T DRAM exhibits a write speed of 2.5 ns, a sense margin of 90 μA/μm, and a retention time of approximately 1 s. Moreover, the energy consumption is 5.0 × 10^−15^ J/bit for the write ‘1’ operation and 0 J/bit for the hold operation. Furthermore, the 1T DRAM shows nondestructive read characteristics, reliable 3 × 3 array operation without any write disturbance, and feasibility in a massive array with an access time of a few nanoseconds.

## 1. Introduction

In recent years, high-density and high-performance memory devices have played a central role in edge computing that has been emerging for innovative technologies such as 5G, the Internet of things, and artificial intelligence [1,2,3]. In particular, one-transistor–one-capacitor (1T–1C) dynamic random-access memory (DRAM) has been improved by reducing the memory cell size with a technology node of the sub-10 nm regime. Nevertheless, the conventional 1T–1C DRAM technology has run up against the limitation of scaling down because the footprint of the capacitor affects the retention time [4,5,6]. Hence, 1T DRAM with a capacitorless structure has attracted attention in DRAM technology [7,8,9,10,11]. Meanwhile, feedback field-effect transistors (FBFETs) with p^+^–n–p–n^+^ silicon nanowires meet the qualifications for 1T DRAM owing to their inherent characteristics such as near-zero subthreshold swings (SSs) and bistable characteristics [11,12,13,14,15,16].

FBFETs operate with the potential barrier modulation via the accumulation and depletion of charge carriers in their channels. When charge carriers are accumulated in their channels, the potential barriers collapse abruptly, and the positive feedback loop is activated in their channels [17]. In the activation of the positive feedback loop, FBFETs are turned on with near-zero SSs. When charge carriers are depleted in their channels, the potential barriers are formed abruptly, and the positive feedback loop is eliminated in their channels [17]. In the elimination of the positive feedback loop, FBFETs are turned off with near-zero SSs. Their near-zero SSs enable a large sensing margin (SM). Moreover, FBFETs exhibit bistable characteristics caused by the presence or absence of a positive feedback loop in their channels, allowing these transistors to operate as 1T memory devices. Their channels can store data without external bias voltages [17]. Accordingly, FBFETs have been extensively studied as 1T memory devices for the application of future memory [17,18,19]. Nevertheless, for their use as DRAM chips with massive array structures, the possible malfunctions in an FBFET array must be evaluated.

There are multiple factors causing the malfunction in the memory chip. First of all, gate oxide reliability is an unavoidable issue in memory devices (including FBFETs) based on the metal–oxide–silicon (MOS) structure. Although the 1T DRAM devices operate at a low field, the gate oxide can break down when the low electric field is applied for a long time, which is called time-dependent dielectric breakdown [20]. Nowadays, the intrinsic breakdown related to various defects is more problematic in an ultra-thin oxide than the extrinsic breakdown related only to the oxide structure. The intrinsic breakdown is caused by the interface trap creation and the gate leakage attributable to the trap-assisted tunneling. However, the intrinsic breakdown is a statistical phenomenon, so it is crucially hard to screen out this possible device failure early during the fabrication stage based on prior research [20]. Moreover, it cannot be improved by device optimization or structural study for the same reason.

The second issue is process-induced device variation. For the nanoscale circuits and systems, the device variation substantially affects the signal system timing and high-frequency characteristics [21]. When millions of the 1T DRAM cells are integrated within a single array, the variation in the cell leads to data corruption during the memory operation. The 1T DRAM array should guarantee at least 98% of cells enable to work despite the device variability for its normal operation [22]. Aggressively scaled FBFETs suffer from random dopant fluctuation, causing device-to-device variation. Moreover, the gate work function fluctuation or surface roughness can induce performance variations in FBFETs [21]. However, recent research has demonstrated that FBFETs exhibit robust stability and that the performance variations may be overcome by the design strategy [23]. Moreover, the performance variations can be substantially suppressed by the improvement in nanoscale fabrication technology [24].

The peripheral circuit performance is another factor influencing the array operation. In the 1T DRAM cell based on FBFETs, two different current levels represent the memory states ‘0’ and ‘1’. To access the information on the 1T DRAM architecture, the selected cell current is compared to a reference current using the peripheral circuit, and the current-mode sense amplifier is used typically in the 1T DRAM architecture [25]. The current-mode sense amplifier operates at a faster speed than the voltage-mode sense amplifier because it directly detects the current difference without waiting for a certain level of the bit line. However, the current-mode sense amplifier is vulnerable to a mismatch between the performances of the component transistors, inducing the error in sensing the memory states. The mismatch can be minimized by improved fabrication technology or by circuitry design solutions.

The aforementioned issues are ably relieved in the array by settling the margin of device variations or by improving the fabrication technology. However, there is a lack of fundamental studies on the disturbance that the newly proposed 1T DRAM receives in a massive array structure. Even if the same random-access memory characteristics are implemented, both the configuration and operating method depend on the device characteristics of an array. For example, the resistive random-access memory (RRAM) requires a selector for a crossbar array owing to the sneak path [26]. Since an FBFET operates as an access transistor and a memory device simultaneously, the cells can be disturbed by the write or read pulses for the near cells through the shared voltage (or current) lines unless reliability is established. Hence, in this study, we investigate the disturbance characteristics of a 3 × 3 1T DRAM array consisting of FBFETs through a mixed-mode technology computer-aided design (TCAD) simulation.

## 2. Device Structure and Simulation Method

Figure 1 shows the schematic design and circuit symbol of a 1T DRAM cell consisting of an n-channel FBFET with a p^+^–n–p–n^+^ silicon nanowire. The dimensional parameters of the 1T DRAM cell included a gated channel length (*L*_G_) of 50 nm, non-gated channel length (*L*_NG_) of 50 nm, channel thickness (*t*_Si_) of 20 nm, and gate oxide SiO_2_ thickness (*t*_ox_) of 5 nm. For each of the regions, a constant doping profile was assumed, and thereby all the p–n junctions were abrupt. The doping concentration was 1 × 10^19^ cm^−3^ in the source and drain regions. The gated region was a p-type doped with a doping concentration of 5 × 10^18^ cm^−3^, and the non-gated region was an n-type doped with a doping concentration of 3 × 10^18^ cm^−3^. An aluminum work function of 4.0 eV was used for the gate, drain, and source electrodes. In a 1T DRAM cell, the voltage pulses of the bit line (BL) and word line (WL) were applied to the source and gate electrodes of the cell, respectively, to determine the memory states, and cell current sensing is performed at the sensing line (SL). In the 1T DRAM cell in the array, the combination of selecting BL and SL determined the specific cell address, and the n-channel metal–oxide–semiconductor field-effect transistor (MOSFET) was used as the SL selector in the mixed-mode simulation.

The 1T DRAM cell can be produced using the fabrication process of a gate-all-around silicon nanowire structure [27]. The intermediate n-doped channel region can be self-aligned with the gate, and the p^+^-doped drain region can be formed via additional masked ion implantation [17]. In a lateral gate-all-around (GAA) silicon nanowire design, the expected cell size is comparable to the traditional DRAM cell (8 F^2^), where F is the minimum feature size [5]. Moreover, the 1T DRAM cell has the potential to be fabricated in a vertical channel [28], and the expected cell size is 4 F^2^.

The simulation in this study was performed with a two-dimensional device structure using a commercial device simulator (Synopsys Sentaurus, O_2018.06) [29]. The two-dimensional structure indicated a cross-sectional view of the 1T DRAM cell so that it was sufficient to consider the electric field on the structural factor. Furthermore, an area factor of 20 nm was specified in the current and charge calculation to determine the device width. The 1T DRAM cell operated with the modulation of the potential barrier by injecting and evacuating charge carriers in the channel region. Accordingly, the 1T DRAM simulation demanded accurate calculations in the band structure, carrier mobility, and recombination. Thus, the high-field saturation, Philips unified mobility, and Lombardi models were used to consider the field and doping dependence of the carrier mobility. Bandgap narrowing (Slotboom model), surface SRH recombination, Auger recombination, Shockley–Read–Hall (SRH) recombination with concentration-dependent lifetimes, and band-to-band tunneling were considered. Moreover, Fermi statistics were applied to perform an accurate simulation. The default parameters were used in the simulation for the models. The transfer characteristic of the 1T DRAM cell was calculated through a transient method for preventing convergence errors at the latch-up voltage. Moreover, the timing diagrams of the 1T DRAM array were simulated using a backward Euler method.

## 3. Results

### 3.1. Characteristics of 1T DRAM Cell

Figure 2a shows the transfer characteristics of a 1T DRAM cell. As *V*_BL_ decreased from −1.1 to −0.8 V, the latch-up voltage (*V*_latch_) shifted to a higher *V*_WL_ from 0.0 to 0.3 V; the decrease in the carrier injection from the source to the gated region was responsible for the *V*_latch_ shift. The bistable characteristics were shown in the transfer curves as *V*_WL_ was swept from −0.5 to 1.0 V and back to −0.5 V. The memory operations of the 1T DRAM cell are shown in Figure 2b. The write, hold, and read operations with a pulse width of 2.5 ns were performed in sequence. To write the ‘0’ state, a *V*_BL_ of −0.4 V and a *V*_WL_ of 0.9 V were applied, and then the positive feedback loop in the channel was eliminated; note that the write ‘0’ operation corresponded to the turning-off of the FBFET comprising the 1T DRAM cell. After the write ‘0’ operation, the *I*_SL_ of the ‘0’ state was negligible under the read operation of *V*_BL_ = −1.0 V and *V*_WL_ = 0.0 V. The write ‘1’ pulse of *V*_BL_ = −1.0 V and *V*_WL_ = 0.3 V activated the positive feedback loop in the channel; note that the write ‘1’ operation corresponded to the turning-on of the FBFET. Consequently, *I*_SL_ reached 1.8 μA under the reading operation, indicating the ‘1’ state. As shown in Figure 2c, under the hold ‘0’ operation, the height of the potential barrier in the gated region was sufficiently high to block the injection of electrons from the source region; thus, the positive feedback loop was hindered in the channel when a *V*_BL_ of −1.0 V was applied. In contrast, for the hold ‘1’ state, the lowering of the height of the potential barrier in the gated region maintained the positive feedback loop in the channel at a *V*_BL_ of −1.0 V. The variation in the ‘0’ and ‘1’ state currents under the reading operation was investigated as a function of the holding time. As shown in Figure 2d, the ‘0’ and ‘1’ states were distinguishable up to an elapsed time of 1 s; the retention time was 1 s. However, after 1 s, the *I*_SL_ of the read ‘0’ operation reached the same current level as the read ‘1’ operation. To understand the retention characteristics of the 1T DRAM cell, we investigated the variation in the hole density of the gated region during the holding ‘0’ and ‘1’ (Figure 2e). The ∆*h*_gated-channel_ represented the difference in the hole density (*h*_gated-channel_) in the gated channel region with the initial hole density (*h*_init_); ∆*h*_gated-channel_ = *h*_init_ − *h*_gated-channel_. Accordingly, the positive (negative) ∆*h*_gated-channel_ referred to the decrease (increase) in the hole density in the gated region. For the hold ‘0’ operation, the ∆*h*_gated-channel_ decreased from ~10^18^ to ~10^16^ cm^−3^ during the elapse of 1 s, and then it dropped drastically after the elapse of 1 s. For the hold ‘1’ operation, the ∆*h*_gated-channel_ varied from −10^17^ to 0 cm^−3^ as a function of the hold time. This result implied that the positive feedback loop was activated when the ∆*h*_gated-channel_ was less than ~10^16^ cm^−3^ and that the read ‘0’ operation failed after the elapse of 1 s (Figure 2d). Moreover, the retention time of the 1T DRAM cell was examined by a full memory cycle with a holding time of 1s (Figure 2f).

### 3.2. Array Characteristics of 1T DRAM

Figure 3a shows the write scheme and disturbing situation for a 3 × 3 array operation. In the 1T DRAM array structure, the read current of the selected cell flowed through the (green) SL selected by the sensing selection line (SSL). When the write operations were performed at the selected cell, the row and column half-selected cells were disturbed by the BL and WL pulses, respectively. For example, when C_11_ was selected, C_10_ and C_12_ were the row half-selected cells (indicated by the blue boxes), and C_01_ and C_21_ were the column half-selected cells (indicated by the red boxes). Figure 3b,c show the timing diagrams of the array when writing ‘0’ and ‘1’ in C_11_, respectively. Before investigating the disturbance, C_01_, C_21_, C_10_, and C_12_ were initialized to be in the ‘0’, ‘1’, ‘0’, and ‘1’ states, respectively. As shown in Figure 3b, to write ‘0’ in C_11_, pulsed voltages of −0.4 and 0.9 V were applied to BL<1> and WL<1>, respectively. After the write ‘0’ operation, a negligible amount of current was detected at SL<1> under the read operation. For C_10_ and C_12_, the read operation revealed that their initial ‘0’ and ‘1’ states were maintained, respectively. Additionally, the initial states of C_01_ and C_21_ were unchanged after the write ‘0’ operation. This demonstrated that the row and column half-selected cells were not disturbed by the write ‘0’ pulse. To write ‘1’ in C_11_, pulsed voltages of −1.0 and 0.3 V were applied to BL<1> and WL<1>, respectively, as shown in Figure 3c. For C_01_, C_21_, C_10_, and C_12_, the read operation confirmed that their initial states were unchanged after the write ‘1’ operation. This demonstrated that the 1T DRAM was capable of memory operation in the array and maintained its memory state even in a disturbing signal.

To examine the reading characteristics during the array operation, the consecutive read pulses were applied to C_00_ and C_01_, which share BL<0> (Figure 4a). The memory states of C_00_ and C_01_ were read in parallel by detecting the currents of SL<0> and SL<1>. For the consecutive read ‘0’ operation, as shown in Figure 4b, the SL<0> current gradually decreased to ~10^−11^ A. Note that the SL<0> current between the current spikes caused by the transition of the bias voltages is shown in this figure. Although the SL<0> current gradually decreased after the read pulses were applied 20 times, the ‘0’ state of C_00_ was stably maintained. Moreover, as shown in Figure 4c, C_01_ maintained a high current level of SL<1> for consecutive read ‘1’ operation. These results revealed that the 1T DRAM array had nondestructive readout characteristics.

Figure 5a shows the parasitic line capacitances in a massive 1T DRAM array structure with numerous cells connected to BL, WL, and SL. To analyze the effect of the line capacitances on the access time, we examined how the propagation delay of the write ‘1’ operation of the 1T DRAM array depended on the parasitic BL, WL, and SL capacitances. The propagation delay was extracted by the time difference between 50% of the input BL (or WL) voltage and 50% of the output SL current. For the parasitic BL and WL capacitances, as shown in Figure 5b,c, the 1T DRAM array exhibited a propagation delay of 0.2 ns regardless of the line capacitance. On the other hand, for the parasitic SL capacitances of 0, 1, 50, and 100 fF, the propagation delays of the 1T DRAM array were 0.2, 0.4, 1.0, and 1.9 ns, respectively (Figure 5d). The result revealed that the access time of the 1T DRAM array was affected only by the parasitic SL capacitance and that it was independent of the parasitic BL and WL capacitances. Furthermore, the parasitic line capacitance was lower than that (40 fF) for 2048 cells at a 10 nm technology node [30]. This implied that the proposed 1T DRAM was feasible for a massive array under the access time of a few nanoseconds.

### 3.3. Performance of 1T DRAM

Table 1 presents a comparison between the proposed 1T DRAM and the recently reported 1T DRAM devices in terms of channel length (*L*_CH_), *L*_G_, SM, I_1_/I_0_, retention time, pulse width, and supply voltage [31,32,33,34,35,36,37]. Regarding the SM and supply voltage, the proposed 1T DRAM was superior to the others; the large SM, approximately ten times larger than that of the other 1T DRAM devices (excluding double-gate (DG) GaAs junctionless transistors (JLTs)), provided the potential to reduce the supply voltage. Regarding the writing/reading speed and retention time, the proposed 1T DRAM was superior or largely equivalent to the others; it exhibited a fast write speed of 2.5 ns and a long retention time of ~1 s.

The gate reliability is a dominant factor determining the endurance for 1T DRAM devices. FBFETs operate with the band modulation mechanism at low voltages, and their storing method is not a burden to the gate oxide. Thus, the proposed 1T DRAMs have infinite endurance if the gate oxide reliability is guaranteed, like the conventional DRAMs. Likewise, the endurance is not an issue in the other 1T DRAM devices (such as Z^2^-FETs and tunneling FETs) operating at low voltages and stored charges in their channel. In contrast, among the 1T DRAM devices based on the floating body effect, the devices using the impact ionization current suffer from the hot-carrier injection into the gate oxide, which meets the limitation in the endurance [7].

The scaling down is still a key factor for footprint-size competitiveness even though the 1T DRAM has an advantage in an area compared to the conventional DRAM. For the use of FBFETs as memory devices, the devices require the positive feedback loop in their channel length (*L*_ch_) so that the positive feedback loop should maintain as the *L*_ch_ is scaled down. Recent research has demonstrated that the *L*_ch_ can be scaled down to ~40 nm [12]. The retention time decreases with the scaling down because of the narrowing of the width of the potential barrier in the channel region. The narrower potential barrier is more vulnerable to unintended carrier injection, such as band-to-band tunneling or leakage from the drain region. Thus, the positive feedback loop is triggered easier, compared to the wider potential barrier, which is unfavorable to the read ‘0’ operation. Meanwhile, the supply voltage and pulse width that trigger the positive feedback loop decrease as the *L*_ch_ is scaled down. Accordingly, not only the energy consumption but also the sensing margin and current ratio decrease because of the reduction in the supply voltage.

Energy consumption is a crucial factor in evaluating memory devices. Table 2 lists the energy consumption of the proposed 1T DRAM calculated by multiplying |*V*_BL_|, *I*_SL_, and the time for each operation. The write ‘1’ and ‘0’ operations consumed 5.0 fJ/bit and 0.1 aJ/bit, respectively, and the read ‘1’ and ‘0’ operations consumed 4.5 fJ/bit and 0.25 aJ/bit, respectively. Moreover, the hold operation retained the data under the zero-bias condition. Therefore, the 1T DRAM could hold ‘1’ and ‘0’ without energy consumption. The energy consumption of the proposed 1T DRAM, the conventional DRAM, and the recently reported 1T DRAMs is compared in Table 3 [32,38,39,40]; the energy consumption of the write and read operations represented the dynamic energy consumption. The proposed 1T DRAM was superior to the conventional DRAM; it exhibited ~2000 times lower energy consumption than the traditional DRAM. Moreover, the proposed 1T DRAM was superior (or comparable) to the other 1T DRAM devices; it exhibited ~100 times lower energy consumption than the other 1T DRAM (excluding double-gate (DG) raised source and drain (RSD) MOSFETs).

## 4. Conclusions

In this study, we demonstrated the DRAM characteristics and reliability of a 1T DRAM array consisting of FBFETs through mixed-mode simulation. An individual cell exhibited a fast write speed of 2.5 ns, a long retention time of ~1 s, and zero energy consumption for holding data. We verified the reliable random-access function in the 3 × 3 array structure without disturbing the write operation in the near cell. Moreover, the 1T DRAM exhibited nondestructive readout characteristics, and the read current was stabilized when a consecutive read pulse was applied. The 1T DRAM array was affected only by the parasitic SL capacitance, and it exhibited an access time of a few nanoseconds (≤1.9 ns), showing its potential application in a massive memory chip. Therefore, these results demonstrated the potential of 1T DRAM for energy-efficient and high-performance memory as a substitute for conventional DRAM.

## Figures and Tables

**Figure 1 micromachines-14-01138-f001:**
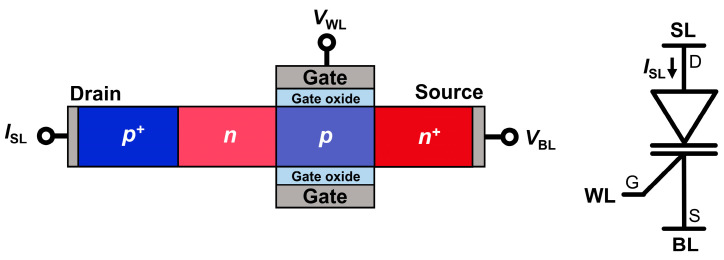
Schematic design and circuit symbol of 1T DRAM cell.

**Figure 2 micromachines-14-01138-f002:**
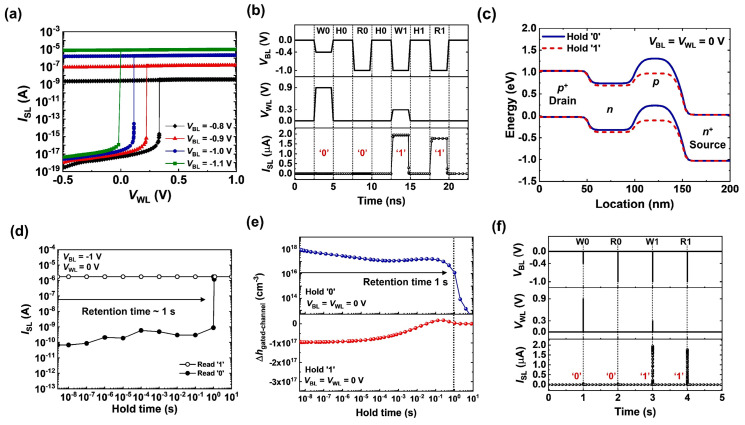
(**a**) Transfer characteristics of 1T DRAM cell. (**b**) Timing diagram of 1T DRAM cell operation. (**c**) Energy-band diagram of 1T DRAM cell under holding ‘0’ and ‘1’ operations. (**d**) Retention characteristics of 1T DRAM cell. (**e**) Variation in the ∆*h*_gated-channel_ of 1T DRAM cell under holding ‘0’ and ‘1’ operation. (∆*h*_gated-channel_ = *h*_init_ − *h*_gated-channel_). (**f**) Timing diagram of 1T DRAM cell with a holding time of 1 s.

**Figure 3 micromachines-14-01138-f003:**
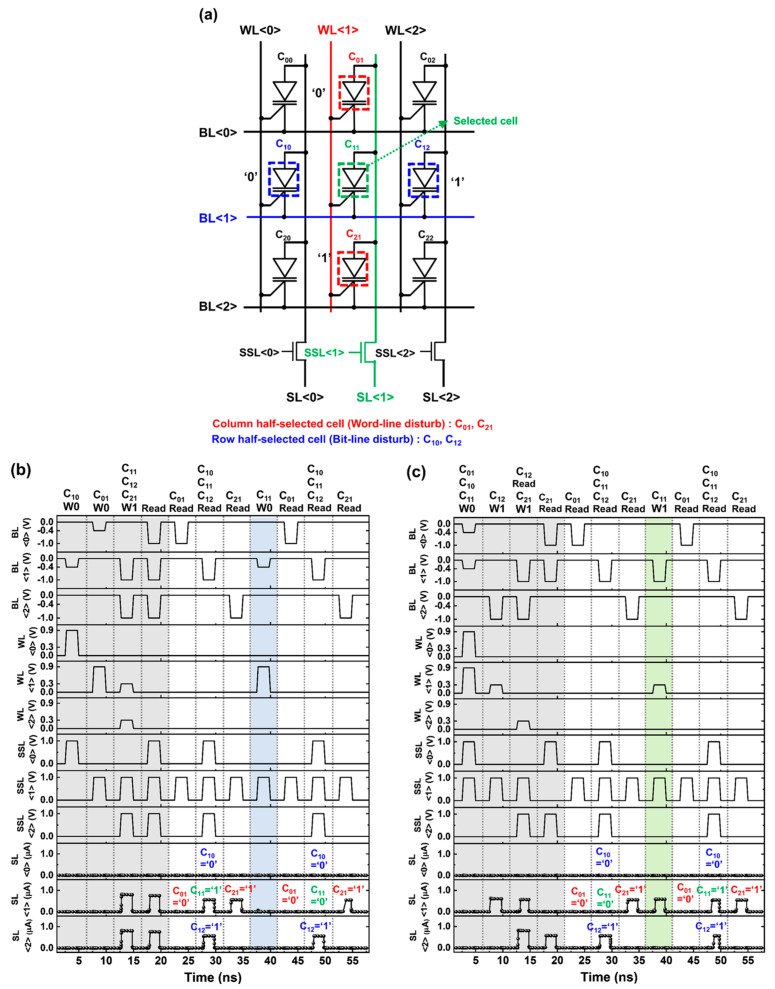
(**a**) Write operation scheme in an array structure with half-selected cell definition. Timing diagrams of the write (**b**) ‘0’ and (**c**) ‘1’ operations for a selected cell in the array structure.

**Figure 4 micromachines-14-01138-f004:**
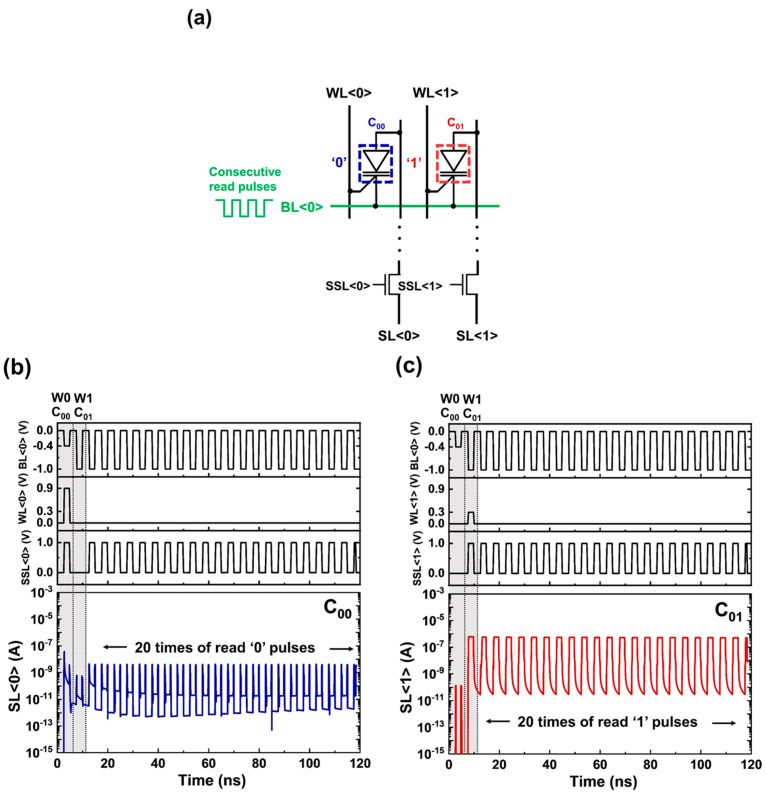
(**a**) Consecutive read operation scheme in an array structure. Timing diagrams of read (**b**) ‘0’ and (**c**) ‘1’ operations with 20 consecutive pulses.

**Figure 5 micromachines-14-01138-f005:**
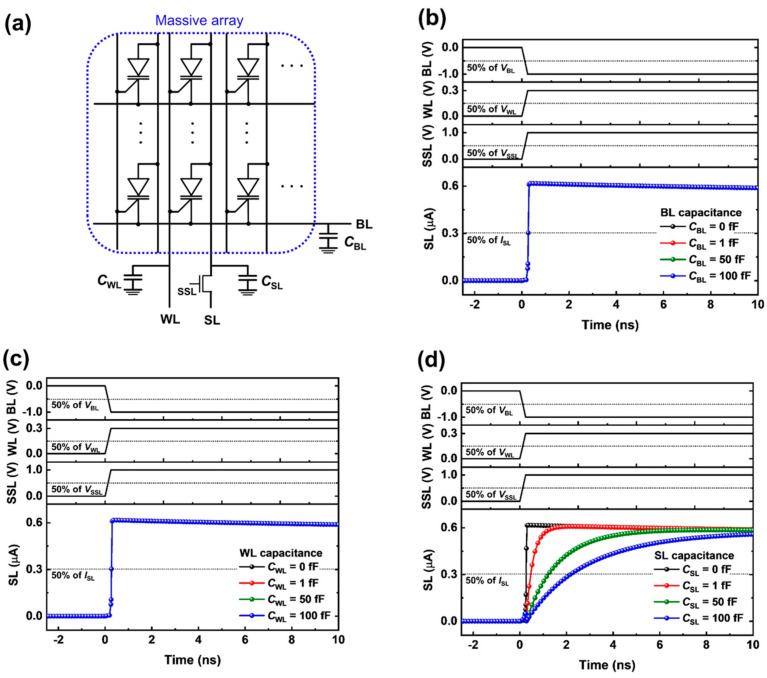
(**a**) Parasitic line capacitance scheme in a massive array structure. Timing diagrams of the propagation delay for the wiring capacitance of (**b**) BL, (**c**) WL, and (**d**) SL.

**Table 1 micromachines-14-01138-t001:** Comparison between the proposed 1T DRAM and the recently published 1T DRAMs.

Device	Ref.	*L*_CH_^1^ (nm)	*L*_G_ (nm)	SM (μA/μm)	Current Ratio (I_1_/I_0_)	Retention Time (ms)	Pulse Width (ns)	Supply Voltage (V)
DG FinFET	[31]	60	60	-	9.4 × 10^5^	0.007	1	2
DG RSD MOSFET	[32]	70	50	1.3	7.5 × 10^2^	330	20	1.5
DG JL 1T DRAM	[33]	120	100	3.5	-	2500	50	1.5
GAA-JLFET	[34]	100	75	0.39	2.23	100	10	1.5
DG GaAs JLT	[35]	100	100	84.4	-	71	10	1.5
L-shaped TFET	[36]	114	50	6.2	1.2 × 10^6^	1700	10	1
Z^2^-FET	[37]	200	100	~20	2.0 × 10^4^	~3	1	1
This study	-	100	50	90	1.8 × 10^4^	~1000	2.5	1

^1^ *L*_CH_ indicates the physical length between the source and drain.

**Table 2 micromachines-14-01138-t002:** Energy consumption of the proposed 1T DRAM.

Operation	|*V*_BL_| (V)	*I*_SL_ (A)	Time (s)	Energy Consumption (J/bit) (*E* = |*V*_BL_| × *I*_SL_ × Time)
Write ‘1’	1.0	2.0 × 10^−6^	2.5 × 10^−9^	5.0 × 10^−15^
Write ‘0’	0.4	~1.0 × 10^−10^		~1.0 × 10^−19^
Read ‘1’	1.0	1.8 × 10^−6^		4.5 × 10^−15^
Read ‘0’	1.0	~1.0 × 10^−10^		~2.5 × 10^−19^
Hold	0.0	--	--	0.0

**Table 3 micromachines-14-01138-t003:** For energy consumption, comparison between the proposed 1T DRAM, the conventional DRAM, and the recently published 1T DRAMs.

Device	Energy Consumption (fJ/bit)	Ref.
Conventional DRAM	> 10,000	[38]
DG RSD MOSFET	2.1	[32]
SiGe QW 1T DRAM	383	[39]
Z^2^-FET	1000~4000	[40]
This work	5.0	-

## Data Availability

Not applicable.

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
