# Peer review of "Disturbance Characteristics of 1T DRAM Arrays Consisting of Feedback Field-Effect Transistors"

_micromachines, 2023, doi:10.3390/mi14061138_

Round 1

Reviewer 1 Report

I have carefully read the work entitled “Disturbance Characteristics of 1T DRAM Arrays Consisting of Feedback Field-Effect Transistors”. In this manuscript, the author reported some important results on the1T DRAM consisting of a FBFET and investigate the memory operation and disturbance in a 3 × 3 array structure through mixed-mode simulations. This work first introduced the characteristics of one single 1T DRAM cell, explained the working principle of FBFET, and demonstrated its retention characteristics. And then, for 3 × 3 DRAM arrays, the author researched on inter cell disturbances and provided evidence for them. Finally, they compared this work with other proposed DRAM performance parameters and showed superiority. According to this manuscript, it was revealed through mixed mode technology computer aided design (TCAD) simulation that the FBFET-based DRAM array does not cause disturbance to the surrounding cells during operation. This study demonstrated the potential of 1T DRAM for high-performance and energy-efficient memory as a substitute for conventional DRAM. Based on these results, I would like to recommend it for publication in Micromachines after addressing the following minor comments.

1.      In Introduction part the author mentioned that "The cell reliability in an array structure is closely related to device malfunctioning". There are multiple factors affecting device reliability, and disturbances in the array are only one aspect. I think the author should clarify why disturbances are studied more clearly.

2.      Line 37 of the first page, there is a number “17”, which I don't quite understand. Is it a reference?

3.      Although multiple parameters have been compared in section 3.3, I still would like to see some comparisons on endurance.

4.      Table 2 is only a comparison of energy consumption under different operating states in this proposed 1T DRAM, so I suggest other device energy consumption comparisons can also be added.

No

Reviewer 2 Report

The technique issue to be solved,

1. It was claimed that the retention time is as long as 1s. However, no simulation or measurement results for maintaining of 1s can be found. Please demonstrate this data for better comprehensions.

2. No layout information can be found here. 1T structure does not equal to compact layout. Please compare this structure with conventional DRAM in terms of layout, linewidth, etc.

good

Reviewer 3 Report

See enclosed file.

Good quality of English language. Minor corrections required. 
